



# Inter-annual variability of wind climates and wind turbine annual energy production

Sara C. Pryor[1], Tristan J. Shepherd[1], Rebecca J. Barthelmie[2]

[1]Department of Earth and Atmospheric Sciences, Cornell University, Ithaca, NY14853, USA
[2]Sibley School of Mechanical and Aerospace Engineering, Cornell University, Ithaca, NY14853, USA

*Correspondence to*: S.C. Pryor (sp2279@cornell.edu)

**Abstract.**

Inter-annual variability (IAV) of expected annual energy production (AEP) from proposed wind farms plays a key role in
dictating project financing. IAV in pre-construction projected AEP and the difference in 50[th] and 90[th] percentile (P50 and
P90) AEP derives in part from variability in wind climates. However, the magnitude of IAV in wind speeds at/close to wind
turbine hub-heights is poorly constrained and maybe overestimated by the 6% standard deviation of annual mean wind
speeds that is widely applied within the wind energy industry. Thus there is a need for improved understanding of the long-
term wind resource and the inter-annual variability therein in order to generate more robust predictions of the financial value
of a wind energy project. Long-term simulations of wind speeds near typical wind turbine hub-heights over the eastern USA
indicate median gross capacity factors (computed using 10-minute wind speeds close to wind turbine hub-heights and the
power curve of the most common wind turbine deployed in the region) that are in good agreement with values derived from
operational wind farms. The IAV of annual mean wind speeds at/near to typical wind turbine hub-heights in these
simulations is lower than is implied by assuming a standard deviation of 6%. Indeed, rather than in 9 in 10 years exhibiting
AEP within 0.9 and 1.1 times the long-term mean AEP, results presented herein indicate that over 90% of the area in the
eastern USA that currently has operating wind turbines simulated AEP lies within 0.94 and 1.06 of the long-term average.
Further, IAV of estimated AEP is not substantially larger than IAV in mean wind speeds. These results indicate it may be
appropriate to reduce the IAV applied to pre-construction AEP estimates to account for variability in wind climates, which
would decrease the cost of capital for wind farm developments.

## 1    Introduction

Wind speeds and thus electrical power production from wind turbines (WT) vary across multiple temporal and spatial scales.
Short-term forecasts (hours to days) of wind speeds at, or near, WT hub-heights (and ideally across the swept area of the WT
rotor) are key to grid management and electricity pricing (Barthelmie et al., 2008;Orwig et al., 2015) and are exhibiting
progressively greater accuracy from direct numerical simulation and statistical post-processing there-from (Pinson et al.,
2007;Sperati et al., 2015;Dowell and Pinson, 2016;Wilczak et al., 2015). Monthly to seasonal forecasts are also increasingly



available to inform planning for WT and grid maintenance (Yu et al., 2015;Torralba et al., 2017). Variability on intra-annual to decadal time scales (Pryor and Barthelmie, 2011;Pryor et al., 2006) arises primarily due to the action of internal climate modes such as El Niño-Southern Oscillation (ENSO) (Schoof and Pryor, 2014;Pryor and Ledolter, 2010;Kirchner-Bossi et al., 2015;Bett et al., 2017;Watts et al., 2017) and climate non-stationarity (e.g. climate change due to the rising concentration

of heat-trapping gases) (Pryor and Barthelmie, 2010;Pryor et al., 2012b;Tobin et al., 2016;Pryor et al., 2012a), and is also key to dictating the electricity produced by WT arrays over their lifetime.

Wind farm developments (i.e. arrays comprising multiple WT) are highly capital intensive with the fuel being free (Lantz et al., 2012). According to some estimates, capital costs (e.g. purchase of wind turbines (WT), installation of foundations and grid connections) comprise up to 80% of the total cost of a typical onshore project over its entire lifetime (Blanco, 2009).

The ratio of capital expenditure to operational expenditures for wind farms in Germany is approximately 0.69 for onshore and 0.54 for offshore wind farms (Steffen, 2018). While the majority (61%) of global 'conventional' power plants are commissioned by state-owned enterprises, private companies commissioned 53% of non-hydro-renewable power plants in 2015 (Steffen, 2018). Further, in Germany, wind farms are overwhelmingly funded through project finance (88% for onshore, 50% for onshore) rather than corporate finance again in contrast to traditional power stations (Steffen, 2018). Thus,

financing risk is particularly important to wind energy (and other non-hydro renewables) and to the levelized cost of energy (LCoE). For a project lifespan of 20 years increasing the cost of capital from 3%/year to 15%/year multiplies the required annual payments by a factor of 2.4 (Krupa and Harvey, 2017).

Cost of capital investments and/or rates of return are determined by the 'risk' associated with each wind energy project, and hence the annual electricity production and variability therein and the resulting anticipated revenue (Feldman and Bolinger,

2016). The variability of revenue due to meteorological/resource variability is described as a specific risk (Gatzert and Kosub, 2016), and requires a minimum debt service coverage ratio if the financing involves debt. Two metrics are often used to quantify viability (and risk) of wind projects in terms of the annual energy production (AEP) (i.e. amount of electricity generated from deployed wind turbines (WT)) over the lifetime of existing/planned wind farms:

- P50: AEP projected to be equalled/exceeded on 50% of years during wind farm operation (P50(AEP)).

- P90: AEP that is associated with a 10% risk of not being reached (P90(AEP)).

Accurate quantification of the wind resource and the P50(AEP) and P90(AEP) presents a significant challenge to current models (Zhang et al., 2015), and even small uncertainties in modelled wind speeds cause major uncertainties in P50(AEP) and P90(AEP) and significantly impact the cost of investment capital in new wind projects (Tindal, 2011;Clifton et al., 2016). Capital investments by the wind energy industry within the United States of America during 2016 are estimated at

\$14.5 billion (Dykes et al., 2017), while estimates of investment in the European offshore wind energy are projected to be between \$90-124 billion over the period 2013-2020 (Gatzert and Kosub, 2016). Even small refinements of perceived and actual project risk deriving from the interannual variability of wind speeds may provide tremendous cost efficiencies (i.e. more accurate assessment of financing costs) and contribute to continuing the recent tendency towards reduced LCoE. It has





been suggested that the LCoE from wind turbines could be reduced by half to $23/megawatt-hour in part due to reductions in financing costs by lowering this long-term production risk (Dykes et al., 2017).

Inter-annual variability (IAV) is used to describe the year-to-year variability in a given property. According to some estimates, IAV contributes "anywhere between 10 % and 25 %" of the overall uncertainty in project energy yield over a 10-
year period (Pullinger et al., 2017). In the wind energy literature IAV is often represented using the standard deviation ($\sigma$) of annual mean wind speeds to the long term mean value and is thus is often quoted as a percentage of the mean. The IAV for annual mean wind speeds (as described using $\sigma$) of 6% is often quoted within the wind energy industry as a representative estimate (Brower, 2012). Indeed the web-site; https://www.wind-energy-the-facts.org/the-annual-variability-of-wind-speed.html states "the annual variability of long-term mean wind speeds at sites across Europe tends to be similar, and can
reasonably be characterised as having a normal distribution with a standard deviation of 6 per cent." This implies that approximately two-thirds of years will have an annual mean wind speed within ± 6 % of the long-term mean. However, much of the research that under-pins this assumption derives from examination of wind speeds at 10-m a.g.l. and employs either data from a limited number of in situ observing stations or relatively coarse resolution reanalysis output (see précis of previous research in Table 1). Further, use of the mean and standard deviation to describe the central tendency and
dispersion of sample implicitly makes an assumption that the sample(s) of annual mean wind speeds are Gaussian distributed. In the event that the sample of annual mean wind speeds is not Gaussian distributed, $\sigma$ is neither a robust or resilient measure of dispersion.

In one of the first published studies on this topic, the IAV of mean wind speeds as described using the $\sigma$ of annual values around the mean across five surface (i.e. within 10 m of the ground) stations in Ireland ranged from 4.7 to 6.4% (Raftery et
al., 1998). In a more recent analysis of surface observations from 16 stations also in Ireland collected over data periods of up to 13 years, $\sigma$ was reported to lie between 4.4 to 6.9% of the mean (Pullinger et al., 2017). Conversely, an analysis of monthly wind speeds at approximately 80-m over the period 1979-2014 from the North American Regional Reanalysis (NARR) data set found 'variations in the wind speed of up to 30%' at some existing wind turbine locations in the United States (Hamlington et al., 2015).
The kinetic energy content of the wind scales with the wind speed cubed. Thus, wind indices (*WI*) that represent the total energy in the flow have also been used in an attempt to better reflect the IAV of the energy available to be harnessed by wind turbines (Table 1). WI are calculated as:

$$WI = \overline{\sum_{j=1}^{n} \frac{U_j^3}{U_{i..k}^3}} \times 100 \qquad\qquad (1)$$

where; j=1 ..n, n = # years, i...k = normalization period, and the mean denotes the spatial average.
The standard deviation of WI integrated over the Scandinavian countries at 10-m height from both NCEP-NCAR and ECMWF reanalyses during 1960-2001 ranged from 8-12% (Pryor et al., 2006). The $\sigma$ of WI for the UK computed using observations collected at 10-m varied from 3.1-7.0% depending on the source, number of stations, data period and whether the data were detrended (Watson et al., 2015). Annual WI generated using Eq (1) are very sensitive to the frequency of





occurrence (and magnitude) of high wind speeds. The actual electrical power derived from wind turbines varies according to the power curves that relate power produced to the wind speed at WT hub-height. This power is zero below cut-in wind speeds, increases rapidly as wind speeds increase and is a constant once the wind speed exceeds that necessary to generate the 'rated power' (RC) (Figure 1) until they exceed a cut-out wind speed (of 25 ms$^{-1}$ for the wind turbine used herein). This

5   non-linearity in turbine power curves means long-term electricity production is typically dominated by the upper percentiles of the wind speed probability density function, but is relatively insensitive to the occurrence of extremely high wind speeds (i.e. above WT cut-out) assuming that they occur only a small fraction of the time (Pryor and Barthelmie, 2010). In short, the IAV in AEP may not be directly proportional to either the IAV of annual mean wind speeds or WI. Very few studies have quantified the actual IAV in wind farm power output. Power output data from a single individual wind farm in the US over

10   the period 2000-2010 ranged between 0.82 and 1.13 of the long-term mean (Wan, 2012). This range in net AEP naturally includes the impact of other factors such as curtailment and maintenance and does not seek to decompose the variability into the root causes.

    Here we investigate IAV in mean wind speeds and WI near typical WT hub-heights using purpose-performed numerical simulations with the Weather Research and Forecasting (WRF) model (v3.8.1). We further estimate IAV in likely AEP due

15   to IAV in wind climates by applying the power curve (Figure 1) from a common wind turbine deployed within the study area to 10-minute wind speed output from these simulations. The results are validated and contextualized using net capacity factors (CF) generated based on power production data from operating wind farms within the simulation domain.



**Table 1: Précis of past research on IAV of wind climates and summary of results presented herein. Results from the current study as shown for grid cells that contain areas with currently operating wind farms denoted by the underlining, and for all other grid cells, and represent results for 90% of grid cells in each class.**

| Descriptor | Data type | Location & # sites | Assumption & Metric | Magnitude | Implied 90% interval of IAV around 'average' value | Reference |
|---|---|---|---|---|---|---|
| Annual mean wind speed | Observations at 10-m a.g.l. | Ireland. 5 stations | Gaussian distribution. σ to describe dispersion | 4.7 to 6.4% | 0.89 to 1.1 | (Raftery et al., 1998) |
| Annual mean wind speed | Observations at 10-m a.g.l. | Approx. 30 (site details not given) | Gaussian distribution. σ to describe dispersion | Approx. 6% | 0.9 to 1.1 | (Raftery et al., 1999) |
| Annual mean wind speed | Observations at 10-m a.g.l. | 16 stations in Ireland (data duration up to 13 years) | Gaussian distribution. σ to describe dispersion | 4.4-6.9% | 0.89 to 1.1 | (Pullinger et al., 2017) |
| Annual mean wind speed and Capacity factors derived from wind speed | Observations at 10-m a.g.l. extrapolated to nominal WT hub-height of between 60-100m and a nominal power curve fitted to generate capacity factors | 6 sites in Scotland (durations of 13 to 43 years) | Dispersion described as difference in X from one year to the next divided by mean. | Δ mean wind speed at 10-m: 10-20% (mean = 15%) Δ mean CF: 11%. | Qualitative remarks imply approx. 0.85-1.15 | (Früh, 2013) |
| Annual mean wind speed | NARR interpolated to 80 m | 1979-2014 | Max % increase or decrease in wind speed anomaly from 35 year mean | Absolute range in different grid cells: 5-40% | N/A | (Hamlington et al., 2015) |
| Annual wind indices | Reanalysis (NCEP/NCAR and ECMWF) 10-m a.g.l. Spatially aggregated country. | 1960-2001 | Gaussian distribution. σ to describe dispersion | 8-12% | 0.80 to 1.2 | (Pryor et al., 2006) |
| Annual wind indices | Spatial composites of 10-m observations, UK | Mostly 29 yr | Gaussian distribution. σ to describe dispersion | 3.1-7% | 0.88-1.15 | (Watson et al., 2015) |
| Annual mean wind speed at approx. 83 m a.g.l. | WRF output at 12 by 12 km grid cells over eastern North America | 2002-2016 | Median and interquartile range | 5.5% 5.2% | 0.95-1.05 0.94-1.06 | This study |
| Annual wind indices at approx. 83 m a.g.l. | WRF output at 12 by 12 km grid cells over eastern North America | 2002-2016 | Median and interquartile range | 14% 11% | 0.85-1.15 0.83-1.17 | This study |
| Annual AEP derived by apply a GE 1.5 MW power curve to 10-min. output | WRF output at 12 by 12 km grid cells over eastern North America | 2002-2016 | Median and interquartile range | 4.9% 5.9% | 0.95-1.05 0.93-1.07 | This study |





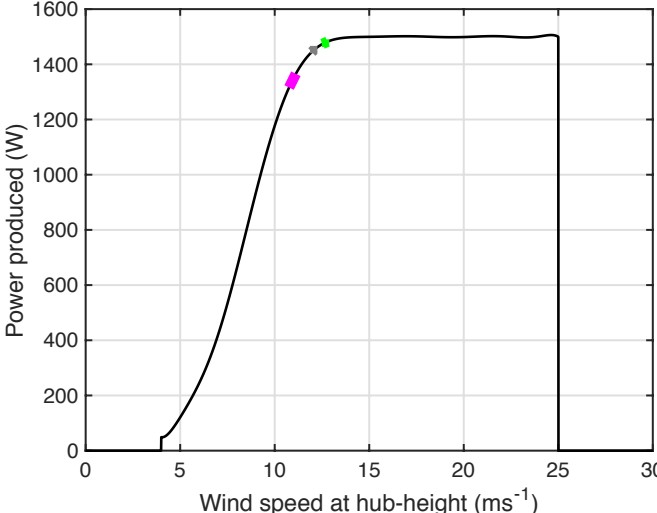

**Figure 1: Power curve (i.e. expected electrical power production as a function of the hub-height inflow wind speed) for the GE 1.5 SLE wind turbine. The three coloured bars shown in magenta, grey and green on this figure show the 95% confidence intervals on the bootstrapped mean annual mean wind speed in the three example grid cells in Texas (TX), Iowa (IA) and New York (NY) state, respectively (see Figure 2 for the locations of these grid cells).**

## 2 Methods

### 2.1 Simulations

Herein we present model-based analyses of the IAV in mean wind speeds, WI and estimated AEP using simulations performed with WRF applied at 12 km resolution over the domain shown in Figure 2. The domain is extended to the west of the region with highest numbers of deployed WT (i.e. the Central Plains) to avoid collocation of the lateral boundaries with a region of strong surface forcing (i.e. the Rocky Mountains). Default settings as specified in the WRF user guide for v3.8 (available at: http://www2.mmm.ucar.edu/wrf/users/docs/user_guide_V3.8/ARWUsersGuideV3.8.pdf) are used for the boundary properties (i.e. five cells are added for boundary value nudging, four of which are in the relaxation zone). Further, a buffer zone comprising 19 grid cells along all four edges of the domain are removed from the simulation output (i.e. used as an adjustment zone to the LBC) prior to the analyses conducted herein. Lateral boundary conditions (LBC) for these simulations are supplied every 6-hours from the ERA-Interim reanalysis data (Dee et al., 2011). The NOAA Real Time Global sea surface temperature (RTG-SST) data set (Gemmill et al., 2007) is used to provide initial SST and Great Lakes conditions and are updated every 24 hours. Data from the 30-arc second Global Multi-resolution Terrain Elevation Data 2010 (GMTED) (Danielson and Gesch, 2011) are used to describe the topography and for consistency with our use of the Noah land surface scheme, land cover is described using the Noah-modified 21-category IGBP-MODIS land use data set (Friedl et al., 2010).



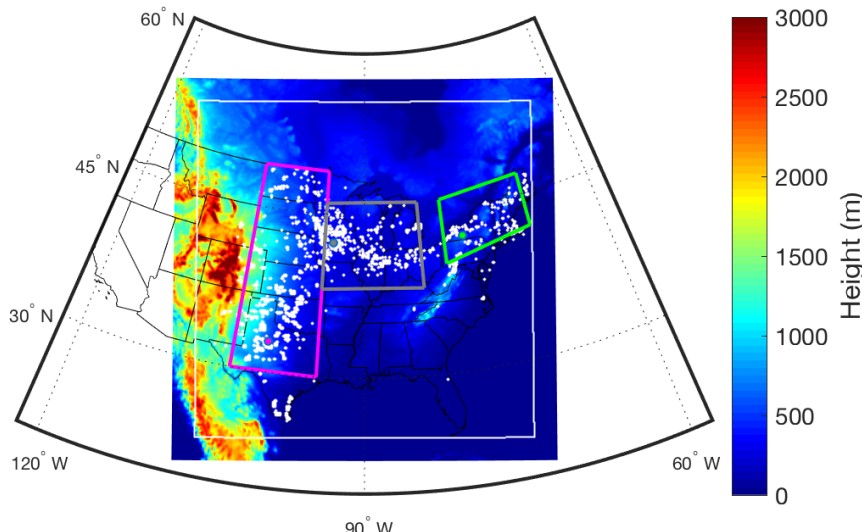

**Figure 2: Simulation domain showing the terrain elevation in each of the total 101761, 12 km by 12 km grid cells. The white box denotes the edge of the adjustment zone applied, and thus delimits the 78961 grid cells (281×281) that are considered herein. The overlaid white dots denote the grid cells in which there were one or more operating WT as of March 2018. The sub domains outlined in magenta, grey and green denote the areas referred to herein as the Central Plains, Midwest, and Northeast respectively. The magenta (TX), grey (IA) and green (NY) dots denote the grid cells used as illustrative examples of the simulated wind climate throughout (e.g. in the bootstrapping of the annual mean AEP and power spectral analyses).**

The time step used for the simulations is 72 seconds, and there are 41 vertical levels (in sigma hydrostatic pressure coordinate) up to a model top at 50 hPa. Eighteen of those levels are below 1 km and the lowest 10 levels represent approximate heights (in flat terrain) of; 16.7, 50.1, 83.6, 117, 151, 184, 218, 253, 293, 338 m a.g.l.. Wind speeds used herein derive from the 3[rd] model layer that represents a height above ground of approximately 83 m (except over the very high/steep complex terrain of the Rocky Mountains in the west of the simulation domain, where the sigma levels are compressed near the surface). The following physics schemes are employed:

- Longwave radiation: 1. Rapid radiative transfer model (RRTM) (Mlawer et al., 1997).

- Shortwave radiation: 1. Dudhia (Dudhia, 1989)

- Microphysics: 5. Eta (Ferrier) (Ferrier et al., 2002).

- Surface-layer physics: 1. MM5 similarity scheme (Beljaars, 1995)

- Land surface physics: 2. Noah land surface model (Tewari et al., 2004).

- Planetary boundary layer: 5. Mellor-Yamada-Nakanishi-Niino 2.5 (Nakanishi and Niino, 2006).



- • Cumulus parameterization: Kain-Fritsch (Kain, 2004).

The simulations start on 15 February 2001 (on the first date for which RTG-SST are available) and run through the end of 31 December 2016. Analyses conducted herein are based on output from 1 March 2011 to 31 December 2016 to allow for a 14 day 'spin-up'. The period required for 'convergence' of inter-annual variability estimates of annual mean wind speed was

previously evaluated by computing the standard deviation of mean annual wind speeds using output from a reanalysis data set of 35 years, and comparing that estimate with the estimate derived from truncated samples there-of. That study found σ converges on the long-term estimate to within +/-15% after 11 years (Pullinger et al., 2017), which implies that the simulations presented herein are of sufficient duration to adequately characterize IAV.

Multiple factors impact the IAV of net AEP from operating wind farms including but not limited to curtailment for system

operation and/or WT maintenance (Clifton et al., 2016), WT wake losses (Clifton et al., 2016;Barthelmie et al., 2013), and wind speed variability. Here we focus on this last factor.

## 2.2 Estimating WI and AEP

Annual mean wind speeds are computed for each grid cell as the arithmetic mean of all 10-minute output from the third model layer in each model grid cell. Wind indices (WI) are computed by applying Eq (1) to the same WRF output and using

a reference time period of 2002-2016. The USGS database of the locations and types of all WT deployed in the continental USA as of March 2018 indicate 57,636 WT were installed in the contiguous USA, of which three-quarters fall within the simulation domain (see Figure 2 for the locations). The most common WT is variants of the GE 1.5 SLE that has a hub-height (HH) of 80 m, a rotor diameter (D) of 77 m and a rated capacity (RC) of 1.5 MW. Thus, WRF output is post-processed to generate a first-order estimate of AEP in each grid cell by assuming there is a single WT deployed in the centre

of each WRF grid cell and applying the power curve of a GE 1.5 SLE (Figure 1) to 10-minute wind speeds from the third model level.

## 2.3 Statistical methods

Output from three example grid cells (located in Texas (TX), Iowa (IA) and New York state (NY), see Figure 2) are used throughout to provide illustrative examples of the simulated wind climate. Time series of 10-minute output from the third

model layer for each calendar year in these grid cells are fitted to Weibull distributions using maximum likelihood methods (Pryor et al., 2004) wherein the probability of a wind speed of a given magnitude is given by:

$$p(U|A,k) = \frac{k}{A}\left(\frac{U}{A}\right)^{k-1} exp\left[-\left(\frac{U}{A}\right)^{k}\right] \qquad (2)$$

Where $A$ is the scale parameter and $k$ is the shape parameter.

The results are used to demonstrate the year-to-year variability in the probability distribution parameters. These time series

from each calendar year are also used with the power curve from the GE 1.5MW WT to generate empirical estimates of the contribution of wind speed bins to the overall estimate power production in each year. Output from these grid cells over the



entire period; January 1 2002 to December 31 2016 are also used to illustrate the temporal scales of variability in the entire sample using fast fourier transform (FFT) applied to compute the variance across a range of frequencies and to present power spectra in the range $f \approx 1 \times 10^{-3}$ to 50 day$^{-1}$. Lastly, time series from these grid cells are also used to consider the question 'how long is long enough?' i.e. what duration of time series is sufficient to characterized the overall annual mean

wind speed and AEP with a certain level of confidence. Time series of the annual mean wind speed and AEP from the three grid cells highlighted in Figure 2 (TX, IA and NY) are subject to a bootstrap analysis (Wilks, 2011) in which the annual mean wind speed and AEP estimates are resampled (with replacement) to generate a synthetic resampled data set of 1000 samples. These are used to compute an estimate of 95% confidence intervals on the long-term mean wind speed AEP and identify the calendar years that differ most profoundly from the bootstrapped mean values in those three locations (Table 2).

**Table 2: Bootstrapped estimates of the mean wind speed in the 3$^{rd}$ model layer and mean Annual Energy Production (AEP), the associated 95% confidence interval (i.e. $\frac{(P97.5(X)-P2.5(X))}{\bar{X}}$) expressed as a percent of the bootstrapped mean values ($\bar{X}$). Also shown are the years that fall furthest from the bootstrapped mean wind speed and AEP values (highest and lowest) for the three grid cells shown in Figure 2. Note: AEP is computed by assuming a single GE 1.5 MW WT is deployed in each 12 km × 12 km grid cell and**

**by applying the power curve of that WT to 10-minute output from the WRF model.**

|  | Bootstrapped mean annual mean wind speed (ms$^{-1}$) | 95% confidence interval (%) | Calendar year (lowest) | Calendar year (highest) | Bootstrapped mean AEP (MWh) | 95% confidence interval (%) | Calendar year (lowest) | Calendar year (highest) |
|---|---|---|---|---|---|---|---|---|
| TX | 10.94 | 3.2 | 2005 | 2008 | 5113 | 3.1 | 2002 | 2012 |
| IA | 12.08 | 2.0 | 2012 | 2007 | 5553 | 2.1 | 2010 | 2006 |
| NY | 12.66 | 2.0 | 2016 | 2009 | 5381 | 2.3 | 2016 | 2010 |

Although it is common practice to describe the IAV of annual wind speeds using a standard deviation around the mean, the assumption that the samples of annual mean wind speed conform to a Gaussian distribution is not always evaluated. The distributions of the 15 values of annual mean wind speed, WI and AEP from each grid cell considered herein are not

normally distributed, rendering the mean and standard deviation poor descriptors of both the central tendency and the dispersion around the central tendency. Indeed, the samples of 15 annual mean wind speed and AEP estimates fail the Anderson-Darling test for normalcy (Wilks, 2011) in 97.7 and 96.3% of grid cells (for a 95% confidence level). Thus, herein we describe the central tendency using the median value (P50) and use the inter-quartile range (IQR, i.e. 25$^{th}$ to 75$^{th}$ percentile range) to describe the dispersion. We also derive estimates of the 90% intervals around the median annual mean

wind speed and AEP (i.e. the range within which 9 out of 10 years are expected to fall), but emphasize these are based on a very small sample size (of 15) and thus are subject to relatively large uncertainty. They are presented solely to permit comparison with 90% intervals around the mean computed using 1.645σ (for normally distributed variables (Wilks, 2011)) applied to past literature that has stated variability in terms of the standard deviation around the mean (Table 1).



P50(AEP) and P90(AEP) are computed for individual calendar years and for rolling consecutive 12-month periods. In the former 10-minute output wind speeds from all grid cells for each of the 15 full calendar years (i.e. 2002, 2003 etc) are subject to the WT power curve and used to compute AEP for each calendar year. In the latter, 10-minute output wind speeds from all grid cells for rolling 12-month periods (i.e. March 2001 to February 2002, April 2001 to March 2002) are subject to

the WT power curve and used to compute AEP for all consecutive 12 month periods. Output from the rolling-12 month periods is used to identify the 12-month period with highest and lowest AEP and those values are evaluated spatially to examine the degree to which that time index and hence the timing of periods with highest and lowest AEP is spatially coherent. The results are considered in the context of monthly indices of the phase of three important internal climate modes that have previously been shown to influence intra-annual and inter-annual variability of wind speeds over the USA (Schoof

and Pryor, 2014;Pryor and Ledolter, 2010); the Pacific North American (PNA) (Leathers et al., 1991), North Atlantic Oscillation (NAO) (Hurrell et al., 2003) and Niño Oceanic Index (ONI) which is a three month running mean of sea surface temperature anomalies in the Niño 3.4 region (Ren and Jin, 2011).

The mean gross capacity factor (CF) for each grid cell is computed as the amount of electrical power produced in each calendar year by applying the power curve for the GE 1.5 MW machine to output from each 10-minute period and

comparing the result to the maximum possible as determined by the rated capacity (1.5 MW) multiplied by the number of hours in a year.

The 1612 of the 281×281 (i.e. 78961) total grid cells (with adjustment zone removed) that contain operating WT as of March 2018 are the primary focus of analyses presented herein (and are referred to as WT grids). Results are also compared to output from the other grid cells (without WT, referred to herein as 'no') to determine whether areas that currently have WT

deployed in them exhibit higher or lower inter-annual variability than typifies the study domain.

### 2.4 Observational data

There are a number of 'bottlenecks' to improved estimation of IAV in mean wind speeds at WT relevant heights and in AEP from WT. These include the lack of publicly accessible high accuracy data at WT relevant heights and high temporal resolution for evaluation of numerical simulations such as those presented herein (Kusiak, 2016). The National Weather

Service (NWS) operates over 900 stations where wind speeds are measured at a height of 10-m a.g.l., but these data are not at or close to WT hub-heights and the actual vertical profile of wind speed is strongly dependent on stability making vertical extrapolation highly uncertain (Badger et al., 2016;Barthelmie et al., 1993;Motta et al., 2005). Additionally, wind speeds as measured by 2-D sonic anemometers at NWS stations are recorded at a resolution of 1 knot ($0.514$ ms$^{-1}$) rounded *up* to the nearest knot when they are archived. The resulting sample is thus systematically biased and pseudo-categorical. Further, in

terms of model validation, local topography and obstacles greatly impact near-surface observations of wind speeds, which makes comparison with grid cell mean values as derived from a numerical model challenging. For these and other reasons, herein we contextualize results of our numerical simulations using observationally derived estimates of the IAV of annual net power production from operating wind farms. Power production data for nearly 1000 operating wind farms as obtained



from the U.S. Energy Information Administration (EIA) (downloaded from https://www.eia.gov/electricity/data/eia923/), are used to monthly capacity factors for each calendar month; January 2001 to December 2016. Sites with ≥ 9 years of data with ≥ 9 months of data availability in each year are used to compute the median annual mean net CF and the normalized IAV therein as represented by the inter-quartile range of annual net CF divided by the median net CF (IQR(CF)/P50(CF)). Sixty-

eight sites meet this data completeness criteria. It is important to note that application of these selection criteria is necessary to ensuring the resulting IQR in CF estimates are robust, but it biases the resulting sample in two important ways; the overwhelming majority of these wind farms are located in Central Plains (Figure 3b) and they tend to represent older generation wind farms in which WT may no longer be under warrantee and may experience declining performance (Olauson et al., 2017), potential leading to inflation of IAV(CF).

**3   Results**

**3.1   Wind speed variability**

Median annual mean wind speed from the 3$^{rd}$ model level (i.e. approx. 83 m a.g.l.) exhibit the expected spatial variability with highest wind speeds over the Central Plains and in a swath across the upper Midwest into the Northeastern states (Figure 3a). This is consistent with the placement of WT in the domain (Figure 2) and previous resource assessments (Pryor

and Barthelmie, 2011;U.S. Department of Energy, 2015;Clifton et al., 2018). Annual gross capacity factors (CF) for grid cells with WT currently deployed in them (WT grid) as derived from the approximations used herein are also consistent with direct observations. The median gross CF computed herein is 40.4% (Figure 3d), which is higher than the net CF derived from the 68 operating wind farms (shown in Figure 3c) of 36%, and slightly lower than the value of 42.5% for WT installations commissioned in 2016 (Wiser and Bolinger, 2017). The median gross CF computed herein for WT grids is

higher than the observed net CF from the 68 operating wind farms, because the net CF also incorporates reductions in power production due to WT maintenance activities, wind turbine wake effects and curtailment for grid management. Observed levels of curtailment over much of the study domain considered herein ≤ 4% during 2007-2012 (Bird et al., 2014). Wind power plant efficiency reductions due to wind turbine wakes are known to be smaller in onshore wind farms than those offshore due to the irregular layouts, higher ambient turbulence intensity and the typically smaller wind turbine densities.

Typical wind turbine induced wakes losses for onshore wind farms are often estimated to be ≤ 5% (Staid et al., 2018), while those offshore are frequently in excess of 10% (Barthelmie et al., 2013). Onshore availability typically exceeds 98% (Carroll et al., 2017), but tends to decrease with WT age (Olauson et al., 2017). Thus gross CF derived from the WRF simulations that assume 100% WT availability (i.e. no downtime for maintenance or curtailment of production), and no wake losses is inevitably higher than the observed values derived from wind farms that have been in operation for more than 10 years. The

estimated median CF derived herein are lower than observed values for new WT deployments in 2016 because the newer WT that are currently being installed have higher WT hub-heights, larger rotors and RC than the GE 1.5 MW WT applied herein.





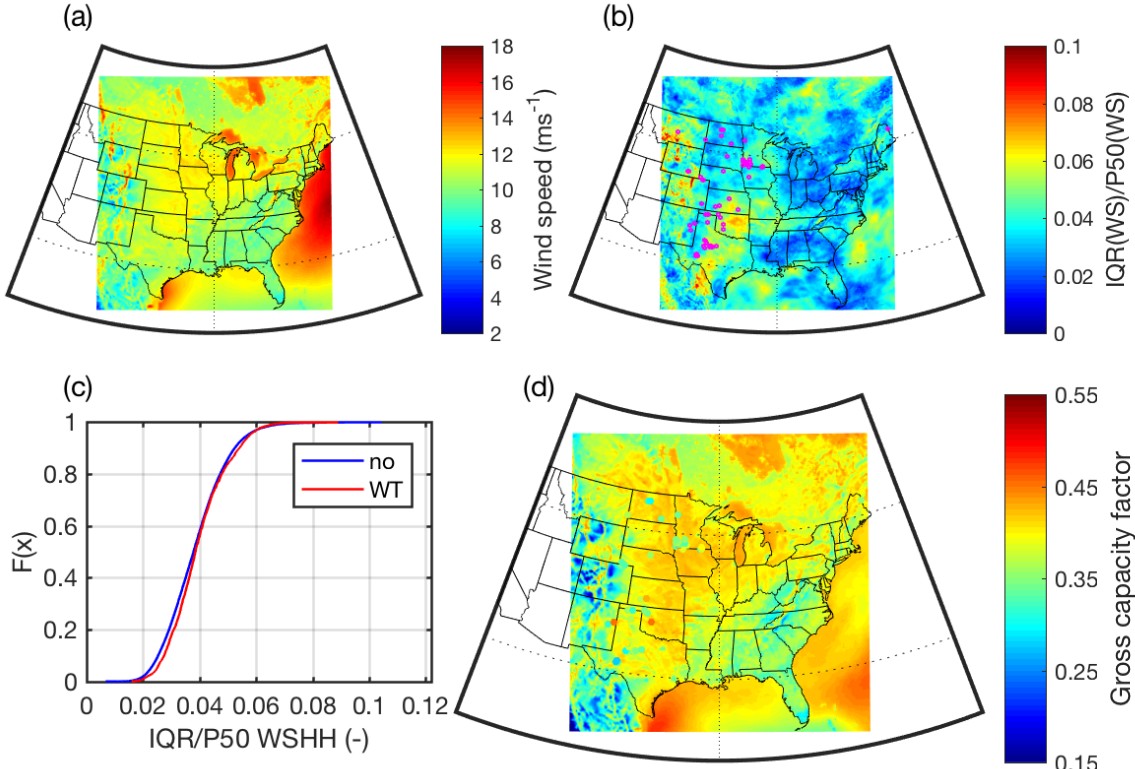

**Figure 3: (a)** Median (i.e. P50) of annual mean wind speeds in the third model layer (approx. 83 m a.g.l) of each 12 km × 12 km grid cell as derived from 10-minute output. **(b)** The normalized inter-quartile range of annual mean wind speeds (IQR(WS)/P50(WS)). The magenta dots shown in this frame denote the locations of operating wind farms from which median CF are shown in frame (d). **(c)** Cumulative density function of (IQR(WS)/P50(WS)) in the sample of grid cells containing WT (shown as WT in the legend) and those that do not (shown as no in the legend). **(d)** Median annual gross capacity factors (CF) for a single WT deployed in each 12 km × 12 km grid cell derived using 10-minute output from the WRF model and the power curve from a GE 1.5 MW WT (see Figure 1). Also shown by the dots in frame (d) are the median net CF computed directly from the power output of operating wind farms. The same colour scale is used for the gross (simulated) and net (observed) CF. If the net and gross capacity factors are equal the wind farm locations (shown in (b)) will not be visible imply agreement between observed and simulated values.

Output for each calendar year from the three grid cells (in Texas (TX), Iowa (IA) and New York (NY), see Figure 2 for locations) conform to two-parameter Weibull distributions as indicated by narrow 95% confidence intervals around the distribution parameters and also illustrate relatively high consistency across the calendar years (Figure 4a). The fraction of power production from each wind speed bin (also plotted in Figure 4a) highlights that the variability of the tail of the wind speed distribution dominates IAV in power production rather than values below the annual mean. Indeed, wind speeds in excess of the annual mean contribute an average of 69, 66 and 57% of the estimated annual total power production in these





grid cells from TX, IA and NY. This emphasizes important potential disconnects between variability of the annual mean wind speed and AEP. The bootstrapped estimate of the annual mean wind speed (and 95% confidence intervals there-on) in the illustrative grid cells from TX, IA and NY state grid fall at a place on the power curve that is relatively close to the wind speed at which the GE 1.5 MW WT generates rated power (i.e. the power output ceases to increase with increasing wind

speeds). This implies that small changes in annual mean wind speeds may not greatly impact the cumulative power output. Analyses of time series of 10-minute output from these three grid cells in the frequency domain indicates that in all of these grid cells the variance is dominated by the meso-$\alpha$ to synoptic time scale ($f \approx 0.2$-$0.5$ day$^{-1}$, thus periods of 2-5 days) (Figure 4b). There is also a clear diurnal peak particularly in IA and NY, while in TX this local maximum is displaced to slightly shorter periods than 1 day. Power spectra derived from output from all three grid cells also exhibit maxima in the frequency

range 2 to $5\times10^{-3}$ day$^{-1}$ (i.e. on annual time scales). This time scale exhibits greatest magnitude of variance in the grid cell from New York state and is of lowest magnitude in Iowa. Variability across all these time scales contributes to the variations in power output from WT, resulting AEP and thus both P50(AEP) and P90(AEP). Figure 4b further reemphasizes the motivation for this research. As shown, the variance at virtually all frequencies considered herein is highest in output from the NY grid cell. This inevitably leads to the question; is a 6% $\sigma$ on annual mean wind speeds and/or AEP appropriate

everywhere?

The normalized IQR of annual mean wind speeds (IQR(WS)/P50(WS)) is < 4% in nearly 60% of WT grid cells, and is < 5% in 83% of WT grid cells and < 6% in 96% of WT grid cells (Figure 3b and c, see summary in Table 1). Recall, a large IQR(X)/P50(X) indicates a site or area with high IAV in parameter X. Thus, this analysis indicates that 5 out of 10 years the annual mean wind speed will fall within ±4% of the long-term average in 90% of the simulation grid cells that contain

operating wind turbines. The estimated 90% confidence interval around the median annual mean wind speed (i.e. 5[th] to 95[th] percentile span in values divided by the median (P50)) is < 8% in half of all WT grid cells, and < 11% in 90% of WT grid cells. Thus, this implies in 9 out of 10 years the annual mean wind speed is expected to fall within ±5.5% of the long-term average in 90% of the simulation grid cells that contain currently operating wind turbines. Comparative estimates of the range of expected annual mean wind speeds derived assuming a Gaussian distribution and a $\sigma$ of 6% are considerably larger.

They yield 90% confidence intervals around of mean that span 19% (i.e. in 9 out of 10 years the annual mean wind speed is expected to fall within ±10% of the long-term average). Several grid cells in the Southern Great Plains indicate higher IQR(WS)/P50(WS) than the median value of 3.8%. However, the lowest 50% of IQR(WS)/P50(WS) of annual mean wind speeds in WT cells is lower than in grid cells without WT. This indicates that, on average, the locations at which WT are currently operating are characterized by lower IAV in wind speeds than typifies the eastern half of North America.





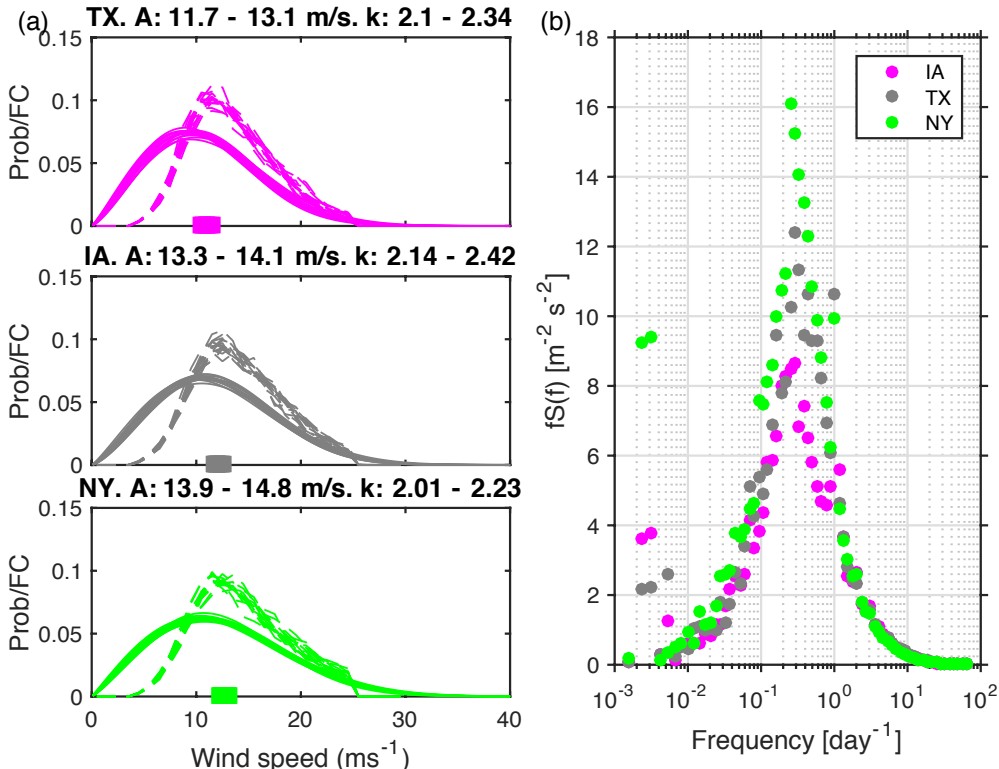

**Figure 4: (a) Weibull distributions calculated for each year for the third model level wind speeds (solid lines) for three example grid cells in Texas (TA), Iowa (IA) and New York (NY) state, respectively (see Figure 2 for the locations of these grid cells). The dashed lines show empirical distributions of the contribution of wind speeds in 1 ms$^{-1}$ bins to the overall annual energy production (FC). The solid coloured boxes on the x-axes indicate the range of mean annual wind speeds in each grid cell. The Weibull parameters for each site are shown above each frame. A is the Weibull scale factor (in ms$^{-1}$) and k is the shape factor. (b) Power spectra of 10-minute disjunct horizontal wind speeds from the third model level from each of these grid cells computed using output every ten minutes for January 1 2002 to 31 December 2016.**

### 3.2    Wind indices and AEP

10    The spatial mean P90(AEP) from WT grid cells is 5157 MWh/yr, while P50(AEP) is 5323 MWh/yr (Figure 5c,d and Figure 6). Comparable figures from grid cells that do not contain the locations of currently operating WT (i.e. no WT grid) are; 4893 and 5078 MWh/yr.





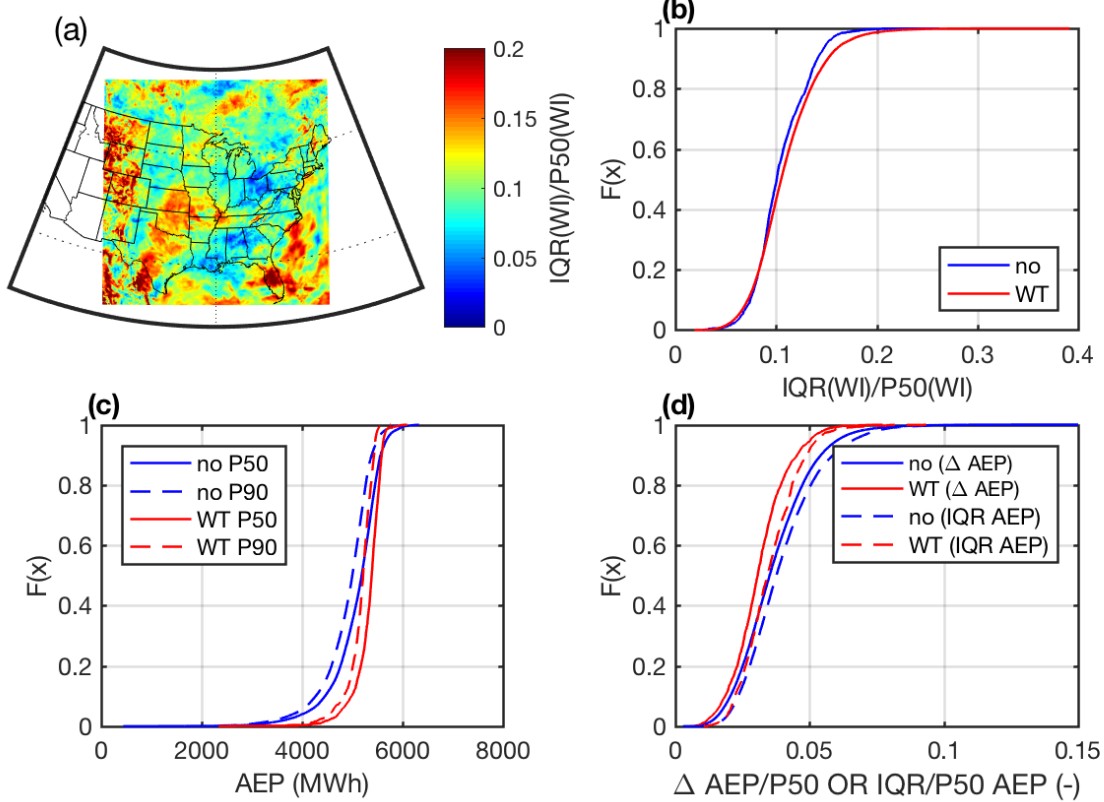

Figure 5: (a) Spatial map of normalized IQR wind index $\frac{P75(WI)-P25(WI)}{P50(WI)}$, and (b) Cumulative density plot of the normalized IQR wind index in sample of grid cells containing WT (WT) and those that do not (no). Cumulative density plots of (c) P50(AEP) and P90(AEP) (in MWh), and (d) normalized difference between AEP P50 and P90 (i.e. $\frac{P50(AEP)-P90(AEP)}{P50(AEP)}$) and IQR(AEP)/P50(AEP) in the sample of grid cells containing WT (WT) and those that do not (no). AEP is computed by assuming a single GE 1.5 MW WT is deployed in each 12 km × 12 km grid cell and by applying the power curve of that WT to 10-minute output from the WRF model.

WI (computed using equation (1)) naturally exhibit larger normalized IQR than annual mean wind speeds (cf. Figure 5a and 3b). Normalized IQR of WI (IQR(WI)/P50(WI)) is < 11% in 60% of WT grid cells, is < 14% in 83% of WT grid cells and is

10    < 15% in 95% of WT grid cells (Figure 5b, see summary in Table 1). However, a similar inflation of IAV is not anticipated for AEP because of the nature of wind turbine power curves (see example in Figure 1). This expectation is realized within the estimated AEP values. The spatial median value of normalized IQR of AEP (i.e IQR(AEP)/P50(AEP)) is 3.4%, thus half of all years are estimated to fall within +/-3.4% of the median (P50) AEP for half of all WT grid cells. The 90% confidence interval tentatively derived as the 5[th] to 95[th] percentile of annual median AEP in each grid cell indicates that for WT grid



cells ranges from 5.0 to 13.5% with a median of 7.9%. Thus in half of all simulation grid cells that cover areas where WT are currently operating, 9 out of 10 years AEP is expected to fall within ±4% of the long-term average (Table 1). Comparative estimates of the range of expected AEP derived assuming a Gaussian distribution and a σ of 6% are considerably larger and yield 90% confidence intervals around of mean that span 19% (i.e. in 9 out of 10 years the AEP is expected to fall within

5    ±10% of the long-term average). Thus, it would appear that assuming a standard deviation (σ) of 6% for the climate-induced inter-annual variability in AEP, is conservative and potentially could be reduced. The normalized difference between P90(AEP) and P50(AEP) in WT grid cells is < 3.1% in 50% of grid cells, and is below 4.6% in 90% of WT grid cells (Figure 5c and d and Figure 6c). Indeed only 1% of WT grid cells exhibit values in excess of 6.4%.

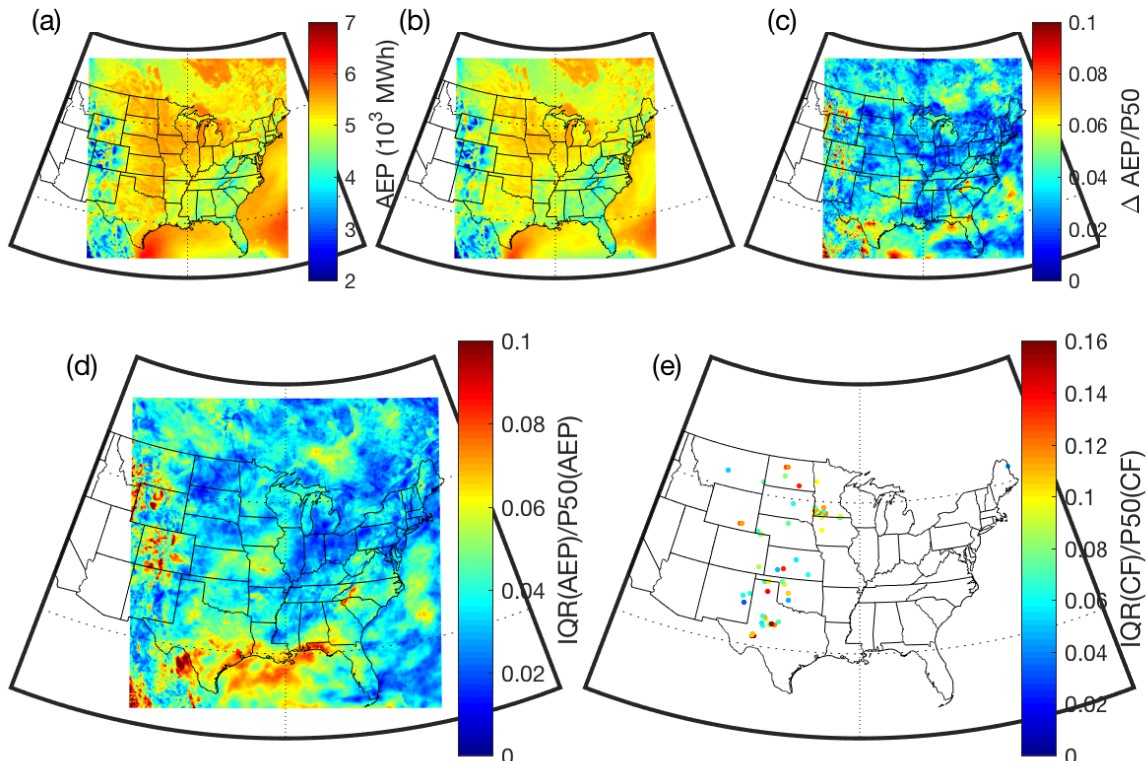

10    **Figure 6: (a) P50(AEP) and (b) P90(AEP) (in $10^3$ MWh) from a single 1.5 MW WT in each 12 km × 12 km grid cell derived using 10-minute output from the WRF model and the power curve from a GE 1.5 MW WT. (c) The difference in P90 and P50 AEP expressed as a fraction of P50 AEP (ΔAEP/P50(AEP)). (d) The normalized interquartile range of AEP (IQR(AEP)/P50(AEP)) in each WRF grid cell and (e) the normalized interquartile range of mean annual capacity factor (IQR(CF)/P50(CF)) from operating wind farms. Note: the scales in frames (d) and (e) differ in order to best depict the full range of values from each data set.**



The mean normalized IQR of gross AEP as derived using output from the WRF simulations and the GE 1.5MW power curve for grid cells containing the 68 operating wind farms considered herein is 3.5% (Figure 6d). The normalized IQR of net CF derived from power production data at these wind farms ranges from 3 to 18%, and has a median value of 9% (Figure 6e). Thus, the power production data from these operating wind farms indicate that in half of them the production during half of all years lies within < ±9% of the long-term average (see summary in Table 1). Our simulations imply that the climate-induced variability at these locations is likely to mean that AEP in half of all years should lie within ±2% of the long-term average, with the remaining variability deriving from other factors such as performance deductions due to WT aging, curtailment, maintenance. These estimates are tentative because the power production data sets are of short duration and containing missing data, and the model simulations are also only 15 years in duration and make a number of assumptions (including use of a single WT power curve). Nevertheless this analysis highlights the need for further studies designed to decompose the IAV of AEP into the root causes of wind climate variability, curtailment, WT availability and WT performance degradation with age.

### 3.3    Scales of coherence in wind speed variability

Understanding the spatial scales of coherence at which wind speed variability on difference timescales is manifest is important to integration of wind energy generated electricity into the grid. Over much of the US, variability of wind speeds on the seasonal to inter-annual timescales is determined by the frequency and tracking of mid-latitude cyclones as dictated by the phase of internal climate modes (Schoof and Pryor, 2014). The timing of the occurrence of the rolling 12-month period of minimum and maximum AEP as computed from the WRF simulations exhibit relatively complex spatial patterns (Figure 7). This indicates that at least at the annual scale, the geographic dispersal of wind turbine deployments is such that it extends beyond regions of high coherence in gross AEP. However, there are also regions of coherence consistent with the importance of large scale climate modes in dictating wind speed anomalies over the contiguous USA (Schoof and Pryor, 2014). Minimum AEP over the upper Midwest (i.e. over Minnesota, Michigan, Illinois, Indiana and Ohio) occurred during 2011 (and early 2012, Figure 7a) during a weak La Niña period (i.e. negative ONI, see Figure 7c), while in the lower Central Plains (Figure 7a) the timing of this minimum was more strongly focused on 2015-2016 during a relatively strong El Niño event (i.e. positive ONI, Figure 7c). Conversely, the upper central Great Plains and parts of the southeast exhibit lowest values for a 12-month period starting in mid-2008 (during a weak la Niña, Figure 7). The timing of maximum AEP is also consistent across the upper Midwest states and is focused on 2007 (Figure 7b). Much of the Central Plains indicate maximum AEP for a period centred on 2011, while estimated AEP the Northeastern states is highest close to the start of the simulation period in 2001/2002. Analysis of the years that differ most from the bootstrapped mean AEP from the three sample grid cells (IA, TX and NY, see Table 2) re-emphasize the findings of the analysis presented in Figure 7. Both indicate that different regions within the eastern USA differ in terms of 12-month period that has lowest AEP and thus when viewed system-wide (i.e. spatially) there are important compensating variations in the wind climate and derived AEP. For example, although 2010 is indicated as a year of relatively low electricity production in Iowa, it is associated with higher





than average AEP from WT in New York state. Figure 7 further illustrates that IAV of AEP (and wind speeds) and the occurrence of higher than normal values is a complex function of the state of multiple climate modes (Schoof and Pryor, 2014). For example, late 2006 saw a weak positive ONI and positive PNA and NAO and was associated with relatively high AEP over much of the Midwest, but late 2009 when ONI was also positive but NAO was negative and PNA was closest to

5      zero was not associated with high estimated AEP over the Midwest.

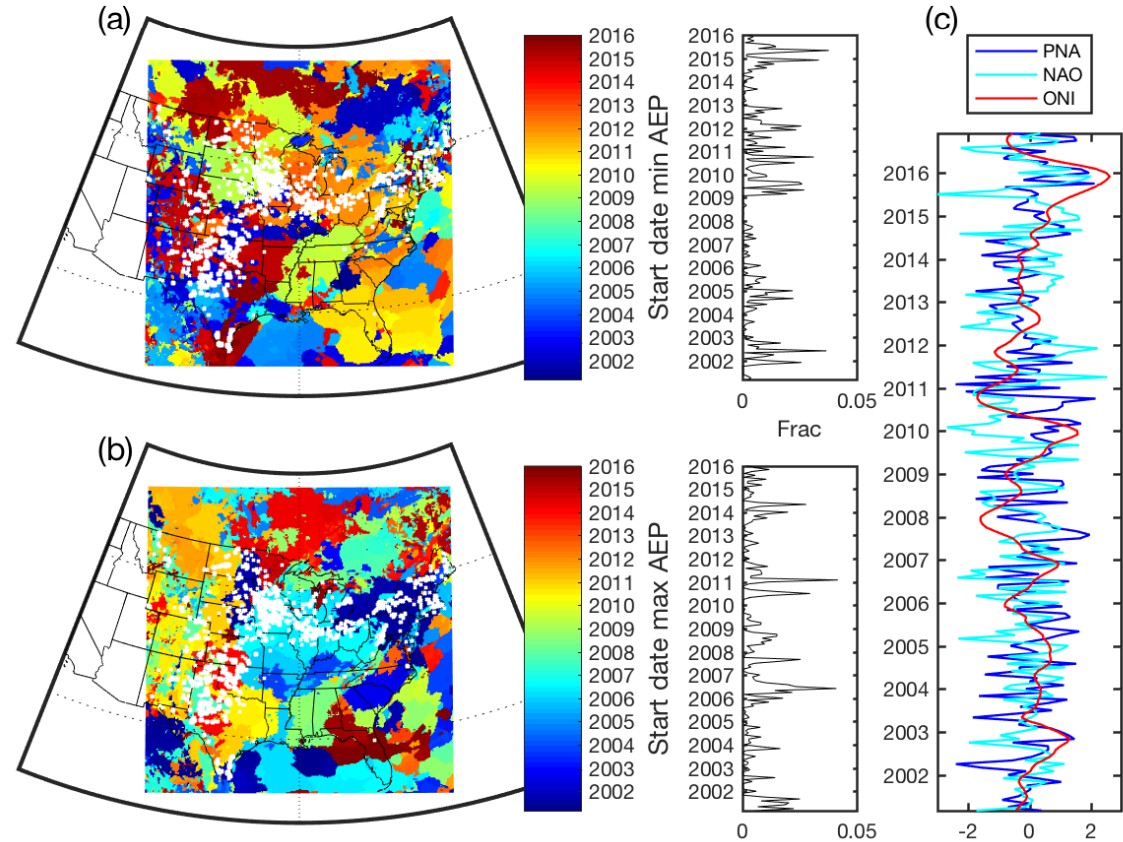

**Figure 7: Timing of the start of the (a) minimum and (b) maximum 12-month rolling AEP value in each 12 km × 12 km grid cell derived using 10-minute output from the WRF model and the power curve from a GE 1.5 MW WT. The white dots indicate the locations of operating WT as of March 2018. As in Figures 5 and 6 AEP is computed by assuming a single GE 1.5 MW WT is deployed in each 12 km × 12 km grid cell and by applying the power curve of that WT to 10-minute output from the WRF model. The panels on the right of each map denote the fraction of all grid cells (Frac) that exhibit a minimum or maximum 12-month rolling AEP in each 12-month period. For a random variable the expectation is this fraction would be 0.0023 in each time period. (c) Monthly indices of the phase of the Pacific North American (PNA), North Atlantic Oscillation (NAO) and Oceanic Niño Index (ONI).**



## 4  Discussion and concluding remarks

This study addresses a key aspect of uncertainty in wind project financing – the magnitude of IAV of wind speeds as manifest in AEP. Over the eastern USA under the contemporary climate the inter-annual variability of annual mean wind speeds close to typical wind turbine hub-heights is smaller than implied by using a standard deviation of 6%. While the IAV

for wind indices is naturally higher than for wind speeds, the IAV of AEP is close to that derived for annual mean wind speeds (see Table 1 and Table 2). The difference between P90(AEP) and P50(AEP) in 12 km × 12 km simulation grid cells that currently contain WT is generally below 5% of P50(AEP), and is < 10% of P50(AEP) for the overwhelming majority of grid cells within the study domain and all grid cells that contain operating WT. Analyses presented herein indicate that AEP in 9 out of 10 years will lie within ±5% of the median value in 90% of grid cells that cover areas that currently contain WT.

Thus, a 6% variability in pre-project estimated mean AEP variability assigned to account for climate variability would appear to be conservative (i.e. likely over-estimates) the 90% confidence interval on AEP over the overwhelming majority of the eastern USA under the contemporary climate.

Although climate modes (such as ENSO) exert an important control on wind regimes over the eastern USA, and coherent sub-domains within the region exist in terms of the timing of the maximum and minimum estimated AEP, these regions of

coherence are sufficiently small that, for example, there are compensating effects between for example, Iowa and New York state. Thus, for a strong and well-connected distribution grid the inter-annual variability in AEP from wind turbines would be small. Indeed for the current distributed WT network the inter-quartile range in system-wide AEP computed from the 15 annual total production estimates derived by equally weighting all grid cells with WT in them is only slightly over 1%.

Naturally, there are a number of caveats that should be applied to our findings. It is implicitly assumed herein that 2001-2016

is a representative climate period. The magnitude of the inter-annual variation in wind speeds, wind indices and AEP reported herein is a function of the simulation period (2001-2016) and the lateral boundary conditions applied (ERA-Interim) to the simulations. It is important to acknowledge that simulated wind climate regimes are a function of the physics packages applied within WRF and the resolution at which the model is applied (Draxl et al., 2014), and further to reiterate that the research presented herein neglects non-climatic factors that influence AEP such as curtailment for system operation and/or

WT maintenance, and IAV in reduced power production efficiency of wind farms (due to wake loss variability resulting from changes in prevailing wind direction). Herein we assume that these effects are secondary to variations in the magnitude of wind speeds. Future work should address the validity of this and the other assumptions employed herein.

This study indicates the urgent need for further research to reduce uncertainty in climate-induced IAV in AEP. Our research suggests the actual IAV in WT generated electricity (AEP) over the eastern USA may be substantially below levels that are

currently adopted in financing mechanisms within the industry. This finding implies that the cost of capital for wind projects may be too high.



## 5    Acknowledgments

This research was funded by the US Department of Energy (DE-SC0016438) and Cornell University's Atkinson Center for a Sustainable Future (ACSF-sp2279-2016). This research was enabled by access to a range of computational resources supported by NSF; ACI-1541215 and those made available via the NSF Extreme Science and Engineering Discovery

Environment (XSEDE) (award TG-ATM170024), and  those of the National Energy Research Scientific Computing Center, a DOE Office of Science User Facility supported by the Office of Science of the U.S. Department of Energy under Contract No. DE-AC02-05CH11231. The authors gratefully acknowledge stimulating conversations with Mr. Ken Westrick and Mr. Ken Davies, the work of Peter Cook in undertaking initial processing of the EIA data and Brandon Barker and Bennett Wineholt for maintaining the Aristotle cloud system. We also gratefully acknowledge the many people who have contributed

to the development of the WRF model.

## 6    Data availability

The USGS Wind Turbine Database is available for download from https://eerscmap.usgs.gov/uswtdb/. Monthly values of the Pacific North American, North Atlantic Oscillation and Niño Oceanic Index are available from the NOAA Climate Prediction Center https://www.esrl.noaa.gov/psd/data/climateindices/list/. Power production data for nearly 1000 operating

wind        farms        are        available        from        the        U.S.        Energy        Information        Administration        (EIA) (https://www.eia.gov/electricity/data/eia923/). Netcdf files containing the derived variables from our WRF output that underpin each of the analyses and figures presented herein have been submitted to the Zenodo repository and will be given a permanent doi and will be available for download once this article is accepted for publication.

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
