# Peer review of "Inter-annual variability of wind climates and wind turbine annual energy production"

_Wind Energy Science, 2018_

## Referee Comment (RC1) · Anonymous Referee #1 · 20 Jul 2018

Synopsis

The ms analyses the inter-annual variability of wind speed and wind turbine energy production over the contiguous United States from WRF simulations and turbine data. They find that the usual assumption of 6 % variability is too high.

The ms deals with an interesting and relevant topic that deserves publication. I suggest publishing the ms after minor revisions according to the points listed below.

Revision items

(1) Figure 4b shows the power spectrum of the simulated wind speeds. It could be interesting to quantify how much the spectrum is deformed on its right-hand side (frequency larger than 1 per day) due to the turbulence parameterisation active in WRF.

[Figure]

At least, a possible influence of the turbulence parameterisation on the high-frequency part of the spectrum should be mentioned.

(2) I wonder whether the authors know the study by Larsén et al. (2016). This study deals with the shape of the wind speed power spectrum and identifies a height dependence of this spectrum. The curves shown in Figure 4b of this ms should be discussed taking into account this height dependency.

(3) Parts of the ms (especially in Section 3) are a bit difficult to read, because the text contains so much abbreviations and percentage numbers. I wonder whether a slight rewording (and maybe the insertion of additional subsections) would help to increase the readability. Not all readers will digest the paper as a whole but usually they would like to pick from parts of it which refer to their needs.

(4) Part of the difficulty to read the paper comes from the fact that a Gaussian statistics (and the "6 %" value stems from such statistics) is compared to a distribution-free statistics throughout the paper. This produces vague statements such as (see, e.g., p.12, lines 10 and 11) "would appear to be conservative". This makes it very difficult for the reader to extract a clear and simple "take-home message".

(5) As a consequence of the item mentioned before, no clear new value is found which could replace the doubted "6 %" value. What would be the new IAV to be applied in future (at least over the contiguous US)?

Reference

Larsén, X.G., S.E. Larsen, E.L. Petersen, 2016: Full-Scale Spectrum of Boundary-Layer Winds. Boundary-Layer Meteorol ., 159, 349–371. DOI 10.1007/s10546-016-0129-x

---

## Referee Comment (RC2) · D. Pullinger (Referee) · 25 Jul 2018

General comments: Thank you for putting this work together, as you say in your paper there is a definite need for more work into this area – and your paper is a valuable contribution to the field. IAV itself has been so poorly represented globally, both in the standard 6% assumption (which is so grossly misused) but also the methodology applied to accounting for IAV. I found your comparison of the different metrics interesting, and it seems to me that there are two main areas for future work (1) how to account for IAV (which methodology) and (2) what values/distributions should be applied in different climates. The overall results for Eastern US are in-line with what I would expect, which is reassuring especially as traditional approaches (assuming normal distributions and wind speed IAV instead or AEP IAV) and the methodology that you have presented

are very different.

Overall the paper was very well written, easy to read and the figures/captions are great.

Specific comments: The first thing that jumped out at me when looking through the report was the reported (and plotted) annual mean wind speed (Table 2 column 2 and again in Figure 3) being above 10 m/s. This seems too high unless there is a different interpretation that I am missing? The NREL US wind speed map at 100m suggests these values should be significantly (about 1/3rd?) lower. Please can you explain why these figures are high (and how this impacts on the results of the work)?

Do you consider the assumption of the generic turbine to have a significant impact on a particular wind farms IAV of AEP? Is this something that you have tested the sensitivity of?

Lastly throughout the report I found it confusing with comparisons made to various different metrics and that these aren't equivalent (P90 is compared with values of "9 in 10 years being within 0.9 and 1.1, and also measures of IQR). Is it possible to convert some of these measures to the same metric (I appreciate the traditional P90 assuming a normal distribution wouldn't work in this case)? That would make it easier for me as the reader to understand the magnitudes of the respective differences.

Technical corrections: Page 1 line 11: should this read "is poorly defined" rather than "poorly constrained"? Page 11 line 2: missing word in "used to *** monthly capacity factors" Page 17 line 14: "different" instead of "difference"

---

## Author Comment (AC1) · 27 Aug 2018

Response to review of Inter-annual variability of wind climates and wind turbine annual energy production

PLEASE NOTE A FULL COPY OF THE MANUSCRIPT WITH TRACKED CHANGES IS ATTACHED AS A SUPPLEMENT TO THIS RESPONSE.

Review 1
Synopsis: The ms analyses the inter-annual variability of wind speed and wind turbine energy production over the contiguous United States from WRF simulations and turbine data. They find that the usual assumption of 6 % variability is too high. The ms deals with an interesting and relevant topic that deserves publication. I suggest publishing the ms after minor revisions according to the points listed below. Response: Thank you for your comments and for this positive assessment.

Revision items (1) Figure 4b shows the power spectrum of the simulated wind speeds. It could be interesting to quantify how much the spectrum is deformed on its right-hand side (frequency larger than 1 per day) due to the turbulence parameterisation active in WRF. At least, a possible influence of the turbulence parameterisation on the high-frequency part of the spectrum should be mentioned. Response: This is correct, use of parameterizations in the mesoscale model and a 12 km grid resolution means high frequency variability is suppressed (linked to point 2)

(2) I wonder whether the authors know the study by Larsén et al. (2016). This study deals with the shape of the wind speed power spectrum and identifies a height dependence of this spectrum. The curves shown in Figure 4b of this ms should be discussed taking into account this height dependency. Response: Yes we are aware of that study. We have added this reference (and their earlier work in 2012) and text to discuss this point in section 3.1.

(3) Parts of the ms (especially in Section 3) are a bit difficult to read, because the text contains so much abbreviations and percentage numbers. I wonder whether a slight rewording (and maybe the insertion of additional subsections) would help to increase the readability. Not all readers will digest the paper as a whole but usually they would like to pick from parts of it which refer to their needs. Response: We have undertaken some rewording throughout the manuscript to hopefully aid readers in following our discussion (see also response to points 4 and 5).

(4) Part of the difficulty to read the paper comes from the fact that a Gaussian statistics

(and the "6 %" value stems from such statistics) is compared to a distribution-free statistics throughout the paper. This produces vague statements such as (see, e.g., p.12, lines 10 and 11) "would appear to be conservative". This makes it very difficult for the reader to extract a clear and simple "take-home message". Response: Yes, it is likely that previous research has invoked parametric statistics because of the ease of interpretation. We have undertaken some rewording to hopefully aid readers in following our discussion (see also response to point 5).

(5) As a consequence of the item mentioned before, no clear new value is found which could replace the doubted "6 %" value. What would be the new IAV to be applied in future (at least over the contiguous US)? Response: This is a very important comment. We have added some text to section 4 that is a tentative recommendation but it is of course offered subject to the caveats we also provide in that section.

Reference Larsén, X.G., S.E. Larsen, E.L. Petersen, 2016: Full-Scale Spectrum of Boundary Layer Winds. Boundary-Layer Meteorol ., 159, 349–371. DOI 10.1007/s10546-016- 0129-x

Please also note the supplement to this comment: https://www.wind-energ-sci-discuss.net/wes-2018-48/wes-2018-48-AC1-supplement.pdf

**Supplement:**

Synopsis: The ms analyses the inter-annual variability of wind speed and wind turbine energy production over the contiguous United States from WRF simulations and turbine data. They find that the usual assumption of 6 % variability is too high. The ms deals with an interesting and relevant topic that deserves publication. I suggest publishing the ms after minor revisions according to the points listed below.
*Response: Thank you for your comments and for this positive assessment.*

Revision items
(1) Figure 4b shows the power spectrum of the simulated wind speeds. It could be interesting to quantify how much the spectrum is deformed on its right-hand side (frequency larger than 1 per day) due to the turbulence parameterisation active in WRF. At least, a possible influence of the turbulence parameterisation on the high-frequency part of the spectrum should be mentioned.
*Response: This is correct, use of parameterizations in the mesoscale model and a 12 km grid resolution means high frequency variability is suppressed (linked to point 2)*

(2) I wonder whether the authors know the study by Larsén et al. (2016). This study deals with the shape of the wind speed power spectrum and identifies a height dependence of this spectrum. The curves shown in Figure 4b of this ms should be discussed taking into account this height dependency.
*Response: Yes we are aware of that study. We have added this reference (and their earlier work in 2012) and text to discuss this point in section 3.1.*

(3) Parts of the ms (especially in Section 3) are a bit difficult to read, because the text contains so much abbreviations and percentage numbers. I wonder whether a slight rewording (and maybe the insertion of additional subsections) would help to increase the readability. Not all readers will digest the paper as a whole but usually they would like to pick from parts of it which refer to their needs.
*Response: We have undertaken some rewording throughout the manuscript to hopefully aid readers in following our discussion (see also response to points 4 and 5).*

(4) Part of the difficulty to read the paper comes from the fact that a Gaussian statistics (and the "6 %" value stems from such statistics) is compared to a distribution-free statistics throughout the paper. This produces vague statements such as (see, e.g., p.12, lines 10 and 11) "would

appear to be conservative". This makes it very difficult for the reader to extract a clear and simple "take-home message".
***Response: Yes, it is likely that previous research has invoked parametric statistics because of the ease of interpretation. We have undertaken some rewording to hopefully aid readers in following our discussion (see also response to point 5).***

(5) As a consequence of the item mentioned before, no clear new value is found which could replace the doubted "6 %" value. What would be the new IAV to be applied in future (at least over the contiguous US)?
***Response: This is a very important comment. We have added some text to section 4 that is a tentative recommendation but it is of course offered subject to the caveats we also provide in that section.***

Reference Larsén, X.G., S.E. Larsen, E.L. Petersen, 2016: Full-Scale Spectrum of Boundary Layer Winds. Boundary-Layer Meteorol ., 159, 349–371. DOI 10.1007/s10546-016- 0129-x

***Response to review of*** **Inter-annual variability of wind climates and wind turbine annual energy production**

**PLEASE NOTE A FULL COPY OF THE MANUSCRIPT WITH TRACKED CHANGES IS ATTACHED AS A SUPPLEMENT TO THIS RESPONSE.**

**Review 2**
General comments: Thank you for putting this work together, as you say in your paper there is a definite need for more work into this area – and your paper is a valuable contribution to the field. IAV itself has been so poorly represented globally, both in the standard 6% assumption (which is so grossly misused) but also the methodology applied to accounting for IAV. I found your comparison of the different metrics interesting, and it seems to me that there are two main areas for future work (1) how to account for IAV (which methodology) and (2) what values/distributions should be applied in different climates. The overall results for Eastern US are in-line with what I would expect, which is reassuring especially as traditional approaches (assuming normal distributions and wind speed IAV instead or AEP IAV) and the methodology that you have presented are very different. Overall the paper was very well written, easy to read and the figures/captions are great.
***Response: Thank you for your comments and for this positive assessment.***

Specific comments:
The first thing that jumped out at me when looking through the report was the reported (and plotted) annual mean wind speed (Table 2 column 2 and again in Figure 3) being above 10 m/s. This seems too high unless there is a different interpretation that I am missing? The NREL US wind speed map at 100m suggests these values should be significantly (about 1/3rd?) lower. Please can you explain why these figures are high (and how this impacts on the results of the work)?
***Response: This is indeed a very important point WRF uses a sigma coordinate system that is approximately terrain following but the height of the 3rd model layer is not uniform across the***

*simulation domain. The mean height above grid cell mean topography varies between 80 and 110 m. We have added a map to show this important point (Figure 2b) and text to section 2.1. The reviewer is quite correct – this may be a source of differences with NREL maps – e.g. the map for 100 m shown at* [https://www.nrel.gov/gis/images/100m_wind/awstwspd100onoff3-1.jpg](https://www.nrel.gov/gis/images/100m_wind/awstwspd100onoff3-1.jpg)*, but of course there are other possible sources; e.g. the 2 km resolution noted on the legend on the NREL map might contribute. Also it is not documented which period was used by AWS Truepower to generate that map (so interannual variability may also be a source of any discrepancies).*

Do you consider the assumption of the generic turbine to have a significant impact on a particular wind farms IAV of AEP? Is this something that you have tested the sensitivity of?
*Response: It might (as we now acknowledge in section 4). We think its probably a much smaller impact than for example vertical extrapolation of wind speeds from 10-m using a power law correction. We consider that IAV should be RELATIVELY insensitive to the power curve – which is for the most commonly deployed WT, but it is certainly worth evaluating in future work.*

Lastly throughout the report I found it confusing with comparisons made to various different metrics and that these aren't equivalent (P90 is compared with values of "9 in 10 years being within 0.9 and 1.1, and also measures of IQR). Is it possible to convert some of these measures to the same metric (I appreciate the traditional P90 assuming a normal distribution wouldn't work in this case)? That would make it easier for me as the reader to understand the magnitudes of the respective differences.
*Response: Yes, it is quite correct (and is a point raised by the other reviewer also). Using a non-parametric description of dispersion is less accessible for some readers. We have undertaken some rewording to hopefully aid readers in following our discussion (see also response to points 4 and 5). See the tracked changed version of the manuscript given below.*

Technical corrections:
Page 1 line 11: should this read "is poorly defined" rather than "poorly constrained"?
*Response: Done*
Page 11 line 2: missing word in "used to *** monthly capacity factors"
*Response: Done (inserted estimate)*
Page 17 line 14: "different" instead of "difference"
*Response: Done*

*Note: One additional change was made to the manuscript – I noticed an error in the legend of Figure 4 that is now corrected.*

[revised manuscript text omitted]